# Peer Power! Secure Peer Attachment Mediates the Effect of Parental Attachment on Depressive Withdrawal of Teenagers

**DOI:** 10.3390/ijerph19074068

**Published:** 2022-03-29

**Authors:** Stefania Muzi, Guyonne Rogier, Cecilia Serena Pace

**Affiliations:** Department of Educational Sciences, University of Genoa, 16129 Genoa, Italy; stefania.muzi@edu.unige.it (S.M.); guyonne.rogier@unige.it (G.R.)

**Keywords:** adolescence, hikikomori, withdrawal, parental divorce, peer attachment, parental attachment, internalizing symptoms, Friends and Family Interview, Youth Self-Report, mediation model

## Abstract

Adolescents’ depressive social withdrawal is a relevant concern for mental health professionals, and it is widespread among community teenagers in form of subclinical symptoms. Different studies suggest that insecure attachment representations increase the adolescents’ likelihood to show symptoms of withdrawal (e.g., loneliness). This study explored the effect of the general attachment internal working model (IWM) and the independent and cumulative effects of the specific attachment representations of parents—in terms of secure base/safe haven—and peers on adolescents’ withdrawal. Additionally, the mediation of peer attachment on the effect of parental attachment on symptoms was explored. All analyses were conducted controlling for the difference between teenagers living with parents together or divorced/separated, as children of divorcees are considered more exposed to stressors. Ninety-one adolescents aged 12–17 years old were assessed online during the COVID pandemic period, employing the Youth Self-Report to assess withdrawal and the Friends and Family Interview to assess attachment-general IWM and attachment-specific representations. Results show no influence of parents together/separated or of the general IWM on withdrawal, but higher parent secure base/safe haven and peer attachment cumulatively predicted 10–21% less withdrawal. Moreover, more positive peer attachment mediated 61% of the effect of the parental secure attachment on withdrawal, revealing an indirect effect of parental attachment on withdrawal through peer attachment. In conclusion, both parents and peers are influential on adolescent mental health, and fostering positive peer relationships can buffer the effect of dysfunctional family relationships on teenagers’ withdrawal.

## 1. Introduction

In recent decades, the social withdrawal of adolescents has been a topic of great interest for practitioners and scholars, due to its worrisome implications on the adolescent’s functioning [1,2,3].

Social withdrawal is defined as a voluntary self-isolation from others, especially peers, which includes deliberately engaging in solitary behavior, not motivated by the desire for solitude [4,5]. It can be expressed through various behaviors, such as avoidance, shyness, or inhibition during social interactions [4], and it is assessed with consensus as an internalizing symptom, indicative of the poorest mental health of adolescents [6].

Indeed, adolescence is a stage where peer relationships play a crucial role in the development and functioning [4], and social withdrawal in this phase is related to lower social-cognitive development [5], less school attendance and achievement [7], lower self-esteem and social confidence [8], and a greater likelihood to display comorbid symptoms or disorders of social anxiety, social phobia, and depression [4,5,9], as well as a higher mortality rate [10]. When a severe social withdrawal lasts more than six months, it can also be indicative of the presence of a syndrome called “Hikikomori” (meaning “withdrawal” in Japanese), referring to people staying at home for long periods without taking part in social or academic/job activities or socializing [11], which is associated with several psychiatric disorders [12]. Additionally, a retrospective study reports greater maladjustment in young adults who were more socially withdrawn when teenagers [13].

Along with the increase in the phenomenon [14], the investigation of correlates and determinants of social withdrawal of adolescents is more and more widespread [5,15]. One trend focuses on the role of significant relationships, highlighting the higher risk of withdrawal symptoms in individuals living in families with a single parent because of separation/divorce or because of a parent’s death or abandonment [16,17], as well as a large consensus in the implication of parent and peer relationships [4,15], but through different mechanisms and with specific roles not yet disentangled.

A similar investigation can be particularly helpful in this period of public health emergency due to the COVID-19 pandemic, contributing to the understanding of why some adolescents forced to be at home with their parents show clinical levels of internalizing symptoms [18], while others show better adjustment [19]. Perhaps the sudden enforced reduction in peer contact, meanwhile with a drastic increase in parental contact, may play a role in adolescent mental health, and withdrawal in particular, in a way that may deserve investigation. Following the literature, particular attention would be reserved for teenagers with divorced parents, and not only because of their suggested likelihood of showing insecure attachment and loneliness [20,21]. Indeed, findings in the literature suggest that the COVID-19 pandemic has been particularly challenging in divorced and separated families, where co-parenting was more difficult and was a source of particular stress and parental conflicts in case of disagreements between parents on teenager’s protection from the virus, management of free time and remote schooling, etc. [22]. Frequent conflicts between parents could have further increased the teenagers’ exposure to stressors, leading to symptomatology, suggesting controlling for the effect of the family structure in this investigation.

### 1.1. Do the Levels of Teenagers’ Withdrawal Vary According to Different Attachment Internal Working Models, Particularly to Preoccupied One?

Attachment theory appears a good theoretical framework for studying the contribution of different significant relationships in the phenomenon of social withdrawal [4,15]. This theory [23] claims that individuals develop strategies for relating to significant people based on meaningful interactions with primary caregivers in infancy. Such strategies are abstracted in mental representations of how to behave and what to expect in significant relationships, called attachment internal working models (IWMs) [23]. The IWMs guide the relational behavior of individuals for their entire life, influencing their social adjustment or vulnerability to psychopathology [24,25].

An infant living with positive parent-child relationships is more likely to develop a secure IWM [23], which helps to flexibly balance connection and independence within significant relationships throughout life, facilitating the individual in building healthy social relationships [23,26]. Instead, in the case of unfavorable parent-child experiences, a child is more likely to develop insecurity in attachment [26]. Insecure IWMs are traditionally categorized into three types [23,27]: insecure-dismissing (or avoidant), where autonomy seeking, independence, and self-reliance prevails; the attachment need of connection is minimized, and there can be a cold diminishment of the importance of significant relationships. Insecure-preoccupied is when seeking proximity and connection with others prevails to the detriment of autonomy and exploration, and there could be anxious hypervigilance to maintain closeness with others, especially parents. Insecure-disorganized is when a child fails in forming a defined strategy of attachment (e.g., in case of parental loss or abuse, or early institutionalization [28]), showing contradictory and disorganized attachment behaviors and expectations, and lacking a mental model guiding relational behavior during life.

Specifically focusing on adolescence, during this stage, the attachment system is re-organized as a consequence of social, relational, cognitive, and neurobiological changes [29,30]. Teenagers’ increasing cognitive abilities allow them to synthesize information from different and compartmentalized relationships’ representations (i.e., with mother, father, teacher, etc.) into a general meta-IWM of attachment, which will be better captured in terms of coherence of attachment narrations from adolescence onwards [27,29]. The links between attachment IWMs and social withdrawal were investigated, revealing both dismissing and preoccupied insecurely attached youths as being more prone to social withdrawal [31,32]. In particular, the majority of studies suggest preoccupation as a form of insecurity more associated with withdrawal [4], being also a predictor of it [31]. Fewer studies showed lower levels of withdrawal in the case of secure IWM, probably because securely attached teenagers show higher social competence and desire for relationality than insecure ones [33], so they have been assumed less likely to withdraw by definition, discouraging more extensive empirical verification despite suggestions by researchers in the field, e.g., [33]. Moreover, there is a clear lack of studies investigating the links between withdrawal and attachment disorganization despite its associations with internalizing symptoms in children and adolescents [34].

### 1.2. Do Higher Parental and Peer Attachment Security Independently Predict Lower Teenagers’ Withdrawal?

Another important change in the attachment system occurring in adolescence is the reorganization of the four functions characterizing attachment relationships—safe haven, secure base, protest for separation, and seek proximity [34,35]. Indeed, studies confirmed that, on the one hand, teenagers’ parents continue to perform the two essential caregiver functions of “safe haven” (SH, i.e., being available and sensitive in comforting and consoling the stressed, troubled, or afraid child), and “secure base”(SB, i.e., supporting the child’s autonomy, encouraging exploration, while also providing practical and emotional help) [35], although in more at-distance ways, in line with teenagers’ greater search for autonomy and independence [36,37]. The capacity of a parent to perform both of these functions, balancing them in response to the child’s needs for closeness or autonomy, is an important contributor to the healthy development of the adolescent [38,39]. Indeed, worldwide longitudinal studies reported lower indicators of social withdrawal (i.e., loneliness, social isolation) in teenagers who perceived parents as more continuously available to perform SH/SB functions [40,41,42,43,44,45], and this has been also confirmed by large cross-sectional studies [39,46], with two studies [29,47] emphasizing the risk for girls. 

On the other hand, during adolescence, functions such as “protest for separation” and “seek proximity” are re-directed from parents towards peers who increasingly become attachment figures, and their representations contribute to the general attachment IWM that will influence adolescents’ mental health [29,36,44,48], including withdrawal. Indeed, two reviews [5,15] conclude that lower peer attachment (measured as lower social competence and worse quality of friendship) is related to more withdrawal in children and adolescents. Coherently, two longitudinal studies [42,44] found a decrease in teenagers’ social withdrawal over the years in the case of higher social competence and more positive friendship quality. Other studies connected lower peer attachment to higher social isolation [45,49], especially in girls.

However, most of the above-mentioned studies utilized exclusively self-report questionnaires, whose answers concerning attachment can be distorted in cases of partially unaware defensive mechanisms typical of teenagers classified as insecure in interviews, such as idealization or anger towards parents [50]. Moreover, withdrawal is measured by indicator rather than a specifically designed measure.

### 1.3. Is Peer Attachment a Mediator in the Relationship between Parental Attachment and Teenagers’ Withdrawal?

In addition to separately investigating the effect of attachment to parents and peers on withdrawal, some authors have also considered the relationships between the two in determining the teenagers’ psychopathological vulnerability [29,39,40,42,44].

Some studies found a cumulative influence of low parent and peer attachment in determining higher internalizing symptoms [29,40] and withdrawal [44] during adolescence. Specifically, two longitudinal studies [42,44] showed that higher security in parent and peer attachment has a cumulative effect in determining a stronger decrease in teenagers’ withdrawal symptoms over the years. Choi et al. [44] also highlighted a stronger positive effect of security in peer attachment than insecurity in parental attachment concerning withdrawal’s decrease.

Additionally, few existing findings documented an interplay between parent and peer attachment in predicting adolescents’ internalizing symptoms [5,29]. For instance, higher security in peer attachment intervenes on the negative effect of parental attachment insecurity on teenagers’ depressive symptoms, particularly in girls [29,51]. Other authors [5,52] reported that teenagers who are insecurely attached to their parents but securely attached to their peers show better adjustment and fewer symptoms [5,42,43,44,52,53].

Despite these few findings, the research gap on the nature of the interplay between parent and peer attachment remains marked, especially concerning this interplay on withdrawal symptoms, where studies are lacking.

The current literature suggests that the quality of early and current parent attachment is likely to predict the quality of peer attachment during adolescence [54,55]. So, it is assumed a double impact of parental insecurity on teenagers’ symptoms, through a direct effect, plus an indirect detrimental effect of parent attachment in increasing insecurity also in peer attachment, which would further aggravate the teenagers’ vulnerability to symptoms [29,56,57]. However, this explanation has never been empirically tested in a study of the mediation (i.e., indirect effect or influence) of parent attachment on a teenager’s withdrawal throughout peer attachment, and never employing specifically designed measures.

### 1.4. The Current Study

In sum, several gaps can be noted in the current literature. First, most of the studies explore internalizing symptoms in general, or solely depression [29], or satellite constructs composing the concept of social withdrawal (e.g., loneliness). Second, concerning the role of IWMs, no studies mentioned disorganized attachment. Moreover, studies on parent and peer attachment are not comprehensive of all facets of the constructs, but mostly of single facets of them (e.g., solely support from parents or peers). Third, all studies exclusively employed self-report questionnaires despite their limits in measuring attachment [34,50], to overcome which an attachment interview was employed in this study for the first time [35]. Fourth, studies considered the separate and cumulative effect of parental and peer attachment, while this is the first study investigating the indirect effect of parent attachment on withdrawal throughout peer attachment.

Therefore, this interview-based study would contribute to filling these research gaps, by involving Italian adolescents, living in intact and separated/divorced families and assessed during the COVID-19 lockdown, to answer the following research questions:RQ1: Do the levels of withdrawal vary according to the different IWMs?RQ2: Are withdrawal levels higher in adolescents with higher levels of insecurity, particularly preoccupation?RQ3: Do higher parental and peer attachment security independently predict lower withdrawal of adolescents?RQ4: Is peer attachment a mediator in the relationship between parental attachment and teenagers’ withdrawal?

## 2. Materials and Methods

### 2.1. Participants and Procedure

Adolescents included in this study participated in the first phase of a longitudinal study conducted in North-West Italy during the first and second waves of the COVID-19 pandemic, from May 2020 to March 2021. Data here included were collected between May and September 2020, when in Italy there were moderate to severe restrictions on social contacts, with intermittent lockdowns of various lengths (two weeks on average).

The entire research obtained approval from the Ethical Committee of the Department of Educational Sciences at the University of Genoa, protocol n. 037.

The total sample of the first phase counted 101 teenagers aged 12–17 years old, enrolled with an age between 12 and 19 years old, and no diagnoses for psychiatric disorders or physical or intellectual disabilities. Of them, 91 (90%) were included in this study according to the criteria of having completed the questionnaire to assess withdrawal and the interview to assess attachment, i.e., IWM, and specific parent and peer attachment representations. There were no differences between the subsample considered in this study and the total sample in age or gender distribution, all *p* > 0.05.

Therefore, 91 teenagers took part in this study and they were aged 12–17 years old (Mean (M) = 14.90, standard deviation (SD) = 1.64; 42% boys), almost all being of Italian nationality (96%) and attending middle (55%) or high school (69%). Teenagers were all Caucasian and belonged mostly to intact families (80% co-living or married parents), while 20% of them came from separated/divorced families.

Almost all participants’ families reported middle-to-high annual income and SES (94%), and the majority of parents were employed (90%), with at least a high-school diploma (55%) or higher educational level (53%). Most of the teenagers had siblings (67%), usually one (62%).

Potential participants were randomly enrolled through public schools. Teenagers interested in voluntarily participating were verbally informed of the aims and procedure of the entire research and about the contents of the informed consent that they were asked to sign to agree to participate. Before the data collection, all the parents signed informed consent as well to agree to the teenagers’ participation. The participants did not receive any incentives for their participation. Because of public health emergency restrictions, data were collected via the internet, in individual sessions on video-calls lasting 1.5 h on average. Participants were asked to respond to an interview and to complete questionnaires remotely.

All the interviews were verbally transcribed to be coded according to official coding guidelines by two certified reliable raters (the first and the third authors). Data of the survey contained in the web repository were periodically downloaded, transposed into a database compatible with the software IBM Statistical Package for the Social Sciences (SPSS) version 24 (IBM Corporation, Armonk, NY, USA), which was saved in hard drives with passwords changed monthly.

### 2.2. Measures

#### 2.2.1. Withdrawal

The participants’ withdrawal was assessed through the syndrome scale withdrawn/depressed of the Youth Self-Report 11–18 years (YSR, [6,58]), a well-known 112-item questionnaire to assess emotional-behavioral symptoms in adolescents. The teenager is asked to rate his/her symptoms by agreeing with a list of sentences on a three-point Likert-type scale (0 = never true, 1 = sometimes true, 2 = most of the time or always true). The score in the withdrawn/depressed scale is the sum of the scores of items 5 (“There is very little that I enjoy”), 54 (“I would rather be alone than with others”), 65 (“I refuse to talk”), 69 (“I am secretive or keep things for myself”), 75 (“I am too shy or timid”), 102 (“I don’t have much energy”), 103 (“I am unhappy, sad or depressed”), and 111 (“I keep from getting involved with others”), with higher scores indicative of more depressive withdrawal. Together with the scales anxious/depressed and somatic complaints, the scale withdrawn/depressed is grouped in the main scale of internalizing symptoms. The version used in this study showed Cronbach’s α between 0.71 and 0.95. In this study, Cronbach’s α was 0.84, reaching the satisfactory level of 0.75 for the withdrawn scale here considered.

#### 2.2.2. Attachment IWMs and Attachment to Parents and Peers

General and specific attachment representations were assessed with the Friends and Family Interview (FFI, [35,59]), an age-adapted interview specifically designed to assess attachment in youths aged 7–17 years old. The youth answers questions about his/her social and school activities and relationships with parents, siblings, and peers, particularly with a best friend decided by the interviewee. The questions are designed to elicit aware and partially unaware aspects of attachment representations. To consider the potentially traumatic impact of eventual losses or adversities due to the COVID-19 pandemic (especially important to rate disorganization), two related questions to the interview in this study, in agreement with the author(s) H. Steele.

The interview was audio- or videotaped to be transcribed verbatim and coded according to a coding system that allowed assigning scores in four scales for the widely known attachment patterns secure-autonomous (F/S), insecure-dismissing (Ds), insecure-preoccupied (E/*p*), and insecure-disorganized, of which the higher score identified the prevalent general attachment IWM. These pattern scores were assigned based on scores (1–4 points) assigned to several scales, including those for specific attachment representations to parents (secure base/safe haven (SB/SH), in three separate scales for mother, father, and another significant figure), and peers (i.e., social competence subscale and two scales for the relationship with the best friend—namely, frequency and quality of contact), plus other scales (narrative coherence, reflective functioning, self-esteem, siblings relationships, affective regulation strategies and differentiation of parental representations). In this study, the general attachment IWM was assessed considering the attachment categories and the scales for attachment patterns, while specific parent and peer attachment representations were assessed through two scales created for this study—namely SB/SH parents (the mean of the scores in the scales SB/SH mother and father) and peer attachment (the mean of the scores in social competence and frequency and quality of best friendship). Two certified raters (the first and second authors) blind double-coded 14% of interviews, and the rest (86%) were coded by one rater. The inter-rater agreement was 94% (k = 0.86) on secure–insecure and four-way classifications, and all scores assigned by the two raters significantly correlated with each other, all *p* < 0.001. For the double-rated interviews, the mean scores of those assigned by the first and the second raters were used as scores. The Cronbach’s α in this study was 0.77 without the computed additional scales here used, remaining satisfactory after their inclusion (α = 0.79)

#### 2.2.3. Demographic Information

Information about participants’ demographics, education and family was collected with a demographic data sheet ad hoc.

### 2.3. Analytic Plan

Data analyses were performed with Statistical Package for the Social Sciences (SPSS) version 24, with PROCESS macro [60]. All analyses were considered statistically significant with *p* < 0.05, and descriptive statistics were fully detailed (frequencies and percentages of attachment categories, M and SD of scores for all measures).

Preliminary scores of boys and girls and teenagers with parents together or divorced/separated were compared through *t*-tests for independent samples, and Pearson’s correlations were performed between participant’s age and all measures’ scores to check the effects of demographics as potential confounding variables, controlled in future correlations if significant.

For the first research question, withdrawal scores of participants classified as secure or insecure in the FFI were compared through a *t*-test for independent samples, and one-way analysis of variance (ANOVA) was performed to check the effect of the four-way attachment classification on YSR/withdrawal scores, which were also correlated with FFI/patterns scales to also check associations at a dimensional level, employing Pearson’s correlation coefficient.

For the second research question, the pattern of Pearson’s correlations between withdrawal and SB/SH parents and peer attachment scores was firstly analyzed, carrying out simple regression models to check the predictive independent or cumulative effects of the two types of specific attachment on withdrawal.

For the third research question, a mediation model was developed through PROCESS macro for SPSS, according to Hayes’ indications [60].

## 3. Results

Detailed descriptive results are reported in Table 1, including the preliminary control for gender and age, which revealed that older participants were more secure in peer attachment, and girls showed higher security in parental attachment, thus demographics were controlled in subsequent analyses.

As shown in Table 1, the preliminary control for the family structure revealed no differences in scores of withdrawal or attachment patterns between teenagers living with together or separated/divorced parents, all *p* > 0.095; therefore, all participants were considered as one unique sample in further analyses.

### 3.1. Differences in Withdrawal According to the Attachment IWM

The distribution of attachment classifications in the FFI was 74 (81.3%) secure and 17 (18.7%) insecure, of which 15 (16.5%) insecure-dismissing, and only 1 participant classified as insecure-preoccupied, and only 1 as insecure-disorganized (1.1% each).

The comparison in withdrawal scores according to the general attachment IWM did not reveal differences between secure (M = 3.46, SD = 2.54) and insecure (M = 3.88, SD = 2.50) participants, *t*(89) = −0.63, *p* = 0.529, or among participants classified as secure-autonomous (M = 3.46, SD = 2.54), insecure-dismissing (M = 3.80, SD = 2.60), insecure-preoccupied (M = 2, SD = 0), or disorganized (M = 5, SD = 0), F(3) = 0.31, *p* = 0.817.

Moreover, there were no correlations between withdrawal and the four attachment patterns, all *p* > 0.067. Therefore, differences in withdrawal according to the prevalent IWM were not found at a categorical or at dimensional level.

### 3.2. Prediction of Withdrawal Based on Parent and Peer Attachment

Withdrawal scores show correlations with both FFI scores of SB/SH parents, r(88) = −0.21, *p* = 0.049, and peer attachment, r(88) = −0.32, *p* = 0.002. Table 2 shows regression models for the prediction of withdrawal based on specific attachment representations, which were all significant. Model 1 and model 2 revealed that both parent and peer attachment security independently predicted less withdrawal, but when considered together, only peer attachment security was a significant predictor, accounting for 8% of the variance in withdrawal.

### 3.3. Mediation Model

A simple mediation analysis was carried out, where the outcome was the score of withdrawal, the parental attachment security in attachment was the predictor, and the peer attachment was the mediator. The mediation model is illustrated in Figure 1, with standardized beta coefficients. The indirect effect of parental attachment on withdrawal was found to be statistically significant (standardized effect = −0.125, 95% CI (−0.237, −0.032)). The difference between the standardized total effect of X on Y and the indirect effect mediated by M was 0.61; therefore, the higher attachment security to parents accounted alone for 39% of the effect in reducing adolescents’ withdrawal symptoms, but 61% of the effect was mediated by the higher attachment security to peers.

## 4. Discussion

In this study, community teenagers were assessed in withdrawal and attachment representations, aiming at identifying pathways of risk connecting them.

Given the likelihood of teenagers showing more loneliness and attachment insecurity following parental separation [20,21] and the additional stress observed in families with separated/divorced parents during the pandemic [22], differences in teenagers’ scores according to the family structure were preliminarily checked, and they were not significant. Therefore, growing up in intact or separated/divorced families did not seem influential on a teenager’s levels of withdrawal or attachment security, so the risk trajectories identified here can be considered valid for all teen participants regardless of their family structure.

The results for the first research question revealed that insecurely attached teenagers did not show more withdrawal than securely attached ones, as there were no differences between teenagers classified as secure or insecure in the age-adapted Friends and Family Interview. Therefore, once following Furlong’s suggestion [33] and the assumption of the lower withdrawal in securely-attached adolescents was tested, this did not prove to be well-founded in this sample.

Moreover, the results answering the second research question did not confirm more withdrawal along with more preoccupation in attachment, given there were no relationships between the levels of social withdrawal and the levels of attachment security, or insecurity in form of dismissal, preoccupation, or disorganization. In addition to contrasting the existing literature [31,32,33], these data provide unpublished information about the apparent irrelevance of attachment disorganization on the symptoms of adolescents’ withdrawal, which can be counterintuitive within current literature [34]. In general, taking together these results seems to suggest the main attachment IWM as not being influential on teenager’s withdrawal, contrasting with the literature [4]. One explanation of this result considers the particular context where the study was carried out, as this is the first study on the topic conducted during the COVID-19 pandemic, in a period of intermittent lock-downs due to the public health emergency’s restrictions. One can assume a reduction in fear of separation for more preoccupied teenagers forced at home with their parents, and this condition could have unexpectedly reinforced significant bonds [61,62,63]. Indeed, despite the concern for the impact of teenagers’ isolation due to the reduction in face-to-face contact [62], for some teenagers lock-downs gave occasion to become closer to significant attachment figures, having more chances to share activities and feelings with their parents [64], to have a virtual meaningful conversation with existing friends, and to seek new friends on social media [61,65]. This condition could have impacted positively both withdrawal and attachment representations, and maybe could have influenced the relationship between the two in a different direction than what was observed before the pandemic [31,32]. Another possible explanation is the low variability in attachment categories in this sample, where more than 80% of teenagers were classified as secure, and almost none of them received a preoccupied or disorganized category, so eventual differences might have been hard to detect with statistical tests. Lastly, together with other authors [29], this result can suggest that the investigation of the impact of the general IWM during adolescence—when it is fluid and under continuous update—could be less informative than studying the specific role of its contributors—i.e., the security in attachment parents and peers.

Indeed, results answering the third and fourth research questions brought new information to the knowledge in the field, for the first time employing specific measures to capture the specific constructs of withdrawal and specific representations of attachment through an interview designed for this scope.

First, once controlling for the effect of gender and age, the results answering the third research question confirmed that attachment security towards both parents and peers reduces the adolescents’ likelihood of showing withdrawal, in line with the literature [29,42,44,46]. As in other studies [29,42,44], these two types of attachment relationships acted independently in determining the risk of a teenager showing withdrawal, similar to what was observed in pathways of risk of other internalizing problems such as depression [29]. Moreover, the models of prediction highlighted that the effect of peer attachment was stronger than the parental one; indeed, only the former was a significant predictor, while the latter lost significance when these contributors were considered together, similar to the finding by Choi et al. [44]. This result supports the idea that, with advancing adolescence, peers progressively become more and more important in the hierarchy of attachment figures, actively contributing to the social well-being of the person [29,30,56,57].

In addition, the results answering the fourth and last research question confirm that attachment to parents and peers not only shows independent and cumulative effects, but they interplay in determining risk trajectories, and the mediation model reported here is an attempt to focus on the nature of this interplay. Specifically, the results reveal an indirect positive effect of parent attachment on withdrawal throughout peer attachment, in line with the theoretical hypothesis made based on the literature on depression [29,40,42,43,44]. This result is particularly outstanding, as no studies before tested this hypothesis, so these are the first empirical data supporting the idea that the relationship between parent attachment and adolescents’ mental health is likely to be due to the impact of parent attachment on peer attachment. Therefore, the wide amount of data documenting the relationships between parent attachment and teenagers’ symptoms may be reread considering that quite a large portion (61%) of the positive or detrimental effect of parental attachment security on these teenager’s symptoms depends on the fact that parental attachment security also influences attachment security to peers, as stated by meta-analyses [54,55]. With this perspective, since the results for the third research question suggest a major role of attachment security to peers in determining withdrawal symptoms, the likelihood of showing security in peer relationships is meanwhile influenced by attachment security to parents, as a primary source of interpersonal confidence for further relationships. In other words, the protective role of parental attachment security on teenagers’ withdrawal seems to be mainly due to the positive influence of parental attachment security on the development of healthy and functional bonds of adolescents with peers. Theoretically, this is in line with the idea that secure attachment to parents in adolescence and adulthood is a distal protective factor for mental health and well-being that fosters its potential throughout the development of attachment bonds with other significant others [18,24,55].

## 5. Conclusions, Limitations, and Future Lines of Research

These outstanding results broaden the current knowledge on the beneficial contribution of peer relationships in adolescence. Extending results on internalizing problems and the more studied depressive symptoms [4,29], this study highlight withdrawal as another possible aspect where parent attachment security can have a positive indirect effect throughout peer attachment. If further studies will support these findings, they would help overcome a deterministic view where attachment to parents alone affects a teenager’s long-term well-being [54,56,57], opening new perspectives that consider its influence on other significant relationships [29]. This broader perspective also allows these results to be read encouragingly by all adults who care for adolescents. Indeed, even if there are compromised situations in the family context, it would be still possible to intervene preventively for the well-being of the adolescent by facilitating relations with peers. Mental health professionals can use specific tools to map attachment in both parent and peer relationships, such as the age-developed interview used in this study [35,59], to design interventions to foster a teenager’s ability to build positive, intimate, and emotionally supportive friendships against the withdrawal, fostering their resilience. As a suggestion, targeted interventions for withdrawn teenagers could be eventually adapted from the evidence-based interventions designed for autism, where social withdrawal is present for different reasons [66]. At a community level, teachers or extracurricular school coaches can propose group activities to promote peer networks, engagement, enthusiasm in social activities, and friendship skills [67].

In one case, this would give time to mental health to simultaneously work in supporting connected and more positive attachment relationships with parents—for instance, through evidence-based short parenting programs [68]. If parents are difficult to engage with, facilitating peer relationships would still be preventative and could increase adolescent resilience [69] and could perhaps facilitate access to mental health services in the event of clinically significant withdrawal symptoms. For instance, researchers and practitioners could develop attachment-oriented interventions aiming to increase the quality of peer relationships, which could be proposed in cases of problematic families or caregivers who are difficult to engage. In this regard, the current pandemic period—characterized by a restriction of school-based and outdoor peer activities—poses a challenge to researchers and professionals to rethink these activities even in an internet-delivered version, to ensure the possibility of prevention even in the riskiest cases of forced closure from external demands, and not only due to adolescent symptoms [70,71].

Of course, the limitations of this study curb enthusiasm, as they reduce the generalizability of the results, which require further research to be considered valid. Indeed, the sample size is small because of the pilot nature of the study and the use of interviews, which has the strength to allow a comprehensive evaluation of attachment representations at an aware and unaware level, but they are longer to administer and code. Further, this is one of few studies that has evaluated withdrawal with a specific measure for assessing this construct and not other satellite ones (e.g., loneliness); however, a single scale of an instrument is not sufficient for obtaining fully reliable results. Moreover, the sample size also affected the results concerning eventual differences due to the family structure, as only 20% of adolescents had divorced/separated parents. This calls for future studies employing larger samples and specific methods of assessment of withdrawal, which so far are lacking when it comes to being based on the evaluation of components of the construct, or on diagnostic criteria proposed for Hikikomori syndrome [68], highlighting a substantial gap in methods if the purpose is to assess sub-threshold symptoms in community adolescents for preventive utility, as in this study.

Lastly, this study on a low-risk community population did not consider the cumulative influence of other potentially influential factors on both attachment and withdrawal, such as the presence of adversities within or outside, such as intimate partner violence, substance abuse in the family, or bullying [72,73], as well the influence of other comorbid symptoms [12]. Future studies could integrate the analyses of these variables to reach a better comprehension of the phenomenon of adolescents’ withdrawal, which this study has confirmed as being still poorly explored, despite being a harbinger of numerous future lines of investigation of clinical relevance.

## Figures and Tables

**Figure 1 ijerph-19-04068-f001:**
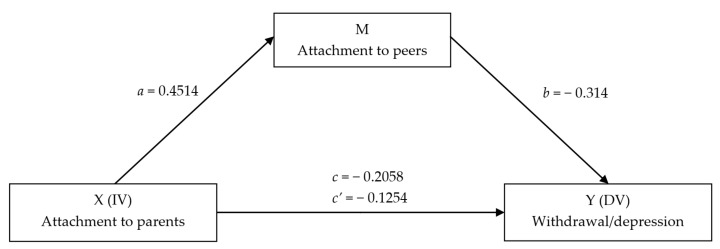
Graphical representation of the mediation model of the effect of the parental attachment security (X) on adolescents’ withdrawal symptoms (Y), as mediated by the security in peer attachment (M).

**Table 1 ijerph-19-04068-t001:** Family structure, age, and gender differences on withdrawal ^a^ and attachment patterns ^b^ toward parents and peers in Italian teenagers.

	Total		Parents				Differences	Relation with Age	Gender				Differences
			Together		Separated				Boys		Girls		
	M	SD	M	SD	M	SD	t(88)	r	M	SD	M	SD	t(89)
Withdrawal	3.55	2.49	3.49	2.54	3.76	2.31	−0.40	−0.01	3.26	2.71	3.70	2.33	−0.82
F/S	3.22	0.77	3.22	0.77	3.18	0.79	0.20	0.12	3.01	0.81	3.37	0.71	−2.20
DS	1.60	0.75	1.62	0.66	1.53	0.72	0.46	−0.08	1.79	0.74	1.46	0.73	2.09
E/*p*	1.16	0.35	1.16	0.36	1.18	0.30	−0.20	−0.19	1.14	0.30	1.17	0.38	−0.34
D	1.06	0.26	1.18	0.50	1.03	0.15	−2.09	−0.06	1.12	0.37	1.02	0.10	1.85
SB/SH parents	2.88	0.71	2.91	0.72	2.72	0.66	0.99	−0.07	2.68	0.75	3.02	0.65	−2.31 *
Peer attachment	3.42	0.54	3.38	0.52	3.61	4.29	−1.69	0.28 **	3.41	0.54	3.43	0.54	−0.21

^1^ N = 91. Boys = 38, girls = 53. ^a^ Youth Self-Report 11–18 years. ^b^ Friends and Family Interview. F/S = secure autonomous, Ds = insecure-dismissing, E/*p* = insecure-preoccupied, D = insecure-disorganize, SB/SH = secure base/safe haven. *p* < 0.05 *, < 0.01 **, and < 0.001 ***.

**Table 2 ijerph-19-04068-t002:** Models of prediction of teenagers’ withdrawal ^a^ based on the predictors: parent and peer attachment ^b^ representations.

	β	SE	95% CI		*p*	*F(1,89)*	R^2^	adj R^2^
**Effect**			LL	UL				
Model 1					0.049	3.94	0.04	0.21
Constant	5.60 ***	1.08	3.45	7.75	<0.001			
SB/SH parents	−0.72 *	0.36	−1.45	0.001	0.049			
Model 2					0.002	9.74	0.31	0.10
Constant	8.77 ***	1.70	5.39	12.15	<0.001			
Peer attachment	−1.53 **	0.49	−2.54	−0.56	0.002			
Model 3					0.008	5.10	0.10	0.08
Intercept	8.98 ***	1.73	5.54	12.42	<0.001			
SB/SH parents	−0.283	0.39	−1.07	0.54	0.479			
Peer attachment	−1.36 *	0.55	−2.45	−0.26	0.016			

^1^ N = 91. Boys = 38, girls = 53. ^a^ Youth Self-Report 11–18 years. ^b^ Friends and Family Interview. SB/SH = secure base/safe haven. *p* < 0.05 *, < 0.01 **, and < 0.001 ***.

## Data Availability

The data presented in this study are available on request from the corresponding author. The data are not publicly available due to institutional policies and because part of a longitudinal project is still ongoing.

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
