# Peer review of "Peer Power! Secure Peer Attachment Mediates the Effect of Parental Attachment on Depressive Withdrawal of Teenagers"

_ijerph, 2022, doi:10.3390/ijerph19074068_

Round 1
Reviewer 1 Report
IJERPH
March 18, 2022
Peer power! Secure peer attachment mediates the effect of pa- 2 rental attachment on depressive withdrawal of teenagers
Introduction
Overall the introduction is well written and argued. I have several minor suggestions to make the claims even stronger.
- The authors state, “When a severe social withdrawal lasts more than six months it can also 45 be indicative of the presence of a syndrome called “Hikikomori” [11], associated with several psychiatric disorders [12]” – please explain Hikikomori, and operationalize this term.
- “Following the literature, particular attention would be reserved to 62 teenagers with divorced parents, and not only because of their suggested likelihood to 63 show insecure attachment and loneliness.” – you need a citation for this claim.
- Typically, attachment theory does not have the word “the” before the title of the theory. Please modify the entire manuscript to make this change.
- This is a personal preference and I will default to the Editor as to if they would like this manuscript modified in this way; please move the research questions throughout the literature review so that the argument of the literature review builds to each RQ.
- Re number your research questions as RQ1, RQ2, RQ3
- RQ1 is actually 2 separate (but related) research questions. Please keep these two RQ together when relocating them in the argument of the literature review, but renumber your questions so that you have a total of 4 research questions.
Materials and Methods
Overall the materials and methods section is very clear. There are a few points that I began to write that the authors needed to add more information, but as I continued through the section the authors clarified every point I had a question about. Nicely done. I rarely have this little input in this area:
- Please report the race/ethnicity of your participants.
- “The participants did not receive 227 any rewards for their participation” – change the word rewards to “incentives.”
Discussion
- Earlier in the manuscript the authors use Research questions, but in the discussion section the authors refer to hypotheses. This manuscript needs to be revised to address this concern.
- The RQ 1-4 need to be inserted into the discussion section (even if they are not addressed in order) in order to make it clearer to the reader that each research question was sufficiently addressed.
- “Indeed, results answering the second and third research questions brought new in- 404 formation to the knowledge in the field, for the first time employing specific measures to 405 capture the specific constructs of withdrawal, and specific representations of attachment 406 through an interview designed for this scope.” – The authors need to revise this portion of the manuscript to address each research question by itself. As it is, neither RQ2 or RQ3 is sufficiently discussed.
Conclusions
- This section is well written and needs very little revision.
- I appreciate the attention to practical and theoretical contributions in this section.
Overall great work on this manuscript. I think with minor revisions this manuscript is nearly ready for publication. Great job.
Author Response
Genoa, March 25th, 2022
To whom it may concern International Journal of Environmental Research and Public Health (IJERPH)
Authors’ responses to Reviewer 1’a comments
We thank the Reviewer 1 for his/her comments. We have attempted to address their suggestions point by point, tracking changes in the manuscript (and coloured them in red). We detailed all changes below:
Reviewer1
Introduction
Overall the introduction is well written and argued. I have several minor suggestions to make the claims even stronger.
- The authors state, “When a severe social withdrawal lasts more than six months it can also 45 be indicative of the presence of a syndrome called “Hikikomori” [11], associated with several psychiatric disorders [12]” – please explain Hikikomori, and operationalize this term.
According with this useful suggestion, we change the statement as follow: “When a severe social withdrawal lasts more than six months it can also be indicative of the presence of a syndrome called “Hikikomori” (meaning “withdrawal” in Japanese), referring to people stay at home for long periods without taking part in social or academic/job activities or socialize [11], which is associated with several psychiatric disorders [12]”
- “Following the literature, particular attention would be reserved to teenagers with divorced parents, and not only because of their suggested likelihood to show insecure attachment and loneliness.” – you need a citation for this claim.
Thanks. Following this advice, we have added two references both in the text and references list:
Çivitci, N., Çivitci, A., & Fiyakali, N. C. (2009). Loneliness and life satisfaction in adolescents with divorced and non-divorced parents. Kuram ve Uygulamada EÄŸitim Bilimleri, 9(2), 513–525.
Crowell, J. A., Treboux, D., & Brockmeyer, S. (2009). Parental divorce and adult children's attachment representations and marital status. Attachment & Human Development, 11(1), 87–101. https://doi.org/10.1080/14616730802500867
Moreover, this check gave us the chance to note an error in the entire manuscript about the number of references in parentheses, so we have corrected the numbers of the references in the entire manuscript.
- Typically, attachment theory does not have the word “the” before the title of the theory. Please modify the entire manuscript to make this change.
Thank for noting this error, we removed “the” before the name of the theory.
- This is a personal preference and I will default to the Editor as to if they would like this manuscript modified in this way; please move the research questions throughout the literature review so that the argument of the literature review builds to each RQ.
We used the research questions in the literature review. After this change, we slightly modified some portions of the text to maintain fluidity. We have maintained three RQs in the introduction, as in our opinion a specific question about preoccupation appeared a little redundant. Waiting for the editor’s preference, we have also left the RQs in the separate section at the bottom of the literature review.
- Re number your research questions as RQ1, RQ2, RQ3
- RQ1 is actually 2 separate (but related) research questions. Please keep these two RQ together when relocating them in the argument of the literature review, but renumber your questions
so that you have a total of 4 research questions.
We renumbered the questions, splitting the first in two.
Materials and Methods
Overall the materials and methods section is very clear. There are a few points that I began to write that the authors needed to add more information, but as I continued through the section the
authors clarified every point I had a question about. Nicely done. I rarely have this little input in this area:
- Please report the race/ethnicity of your participants.
We thank the Reviewer1 for this comment. We have added “Teenagers were all born in Italy and Caucasian”.
- “The participants did not receive any rewards for their participation” – change the word rewards to “incentives.”
We modified the word as suggested.
Discussion
- Earlier in the manuscript the authors use Research questions, but in the discussion section the authors refer to hypotheses. This manuscript needs to be revised to address this concern.
We removed references to hypotheses in the entire manuscript, referring to “answers” to the research questions
- The RQ 1-4 need to be inserted into the discussion section (even if they are not addressed in order) in order to make it clearer to the reader that each research question was sufficiently addressed.
- “Indeed, results answering the second and third research questions brought new information to the knowledge in the field, for the first time employing specific measures to capture the specific constructs of withdrawal, and specific representations of attachment through an interview
designed for this scope.” – The authors need to revise this portion of the manuscript to address each research question by itself. As it is, neither RQ2 or RQ3 is sufficiently discussed.
We thank Reviewer1 for these observations. We re-ordered the discussion for more clarity, highlighting the referral to each of the four research questions, and enriching the discussion of RQ3 (previously RQ2) and RQ4 (previously RQ3):
e.g. “the results answering the third research question confirmed that attachment security towards both parents and peers reduces the adolescents' likelihood to show withdrawal, in line with the literature [29, 42, 44, 46]. Like in other studies [29, 42, 44] these two types of attachment relationships act independently in determining the risk of a teenager showing withdrawal, similarly to what was observed in pathways of risk of other internalizing problems such as depression [29].”
Or “In addition, the results answering the fourth and last research question confirm that attachment to parents and peers do not only show independent and cumulative effects, but they interplay in determining risk trajectories, and the mediation model reported here is an attempt to focus on the nature of this interplay. […] Therefore, the wide amount of data documenting the relationships between parent attachment and teenagers’ symptoms may be reread considering that a quite large portion (61%) of the positive or detrimental effect of parental attachment security on these teenager’s symptoms depends on the fact that parental attachment security influences the attachment security also to peers, as stated by meta-analyses [54, 55]. With this perspective, if the results for the third research question suggest a major role of attachment security to peers in determining withdrawal symptoms, meanwhile the likelihood to show security in peer relationships is influenced by the attachment security to parents, as a primary source of interpersonal confidence for further relationships. In other words, the protective role of parental attachment security on teenagers’ withdrawal seems to be mainly due to the positive influence of parental attachment security on the development of healthy and functional bonds of adolescents with peers.”
Conclusions
This section is well written and needs very little revision. I appreciate the attention to practical and theoretical contributions in this section. Overall great work on this manuscript. I think with minor revisions this manuscript is nearly ready for publication. Great job.
We really thank Reviewer1 for his/her encouraging comment.
Kind regards
The Authors

Reviewer 2 Report
Dear authors, thank you very much for your effort and research in this interesting topic. I am grateful for your contribution.
Introduction:
Line 113 -typ error "safe have" Lines 138-140: "However, most of the above-mentioned studies utilized exclusively self-report questionnaires, whose answers can be distorted in specific cases [53], and withdrawal is measured by indicator rather than a specifically designed measure" - what did the authors infere here? were the answers distorced? Please be very clear in your statments, do not leave any uncleared inference on reviews and resutls. If this really happened than just change the sentence.
Materials and Methods:
Authors declared to have used SPSS Version 54, this might be an error since the actual version is 28. A guide / cut-off for the Pearson correlation coefficient is missing. Please inform the reader how to interpret these values. Lines 214-215: authors described the sample and affirm that the gender distribution is 54% boys, but at Table 1 we can see that the frequency of boys in the sample is actually 42% (38 out of 91) - where is the missing piece?
In generally: authors comment on the signification of comparisons based on p-values being higher than different cut-offs, e.g. line 319 p > 0.095 or line 340 p > 0.067 - what are those values? Usually one determines a significance level, say 0.05 and report the significane according to that. Is it possible to display (Figure 1) standardized beta coefficients for the models?
How are the readers supposed to interpret the a=0.3251 and b=1.534 if they are not at the same scale?
Author Response
Genoa, March 25th, 2022
To whom it may concern International Journal of Environmental Research and Public Health (IJERPH)
Authors’ responses to Reviewer 2’s comments
We thank the Reviewer 2 for his/her comments. We have attempted to address their suggestions point by point, tracking changes in the manuscript (and coloured them in red). We detailed all changes below:
Reviewer2
Dear authors, thank you very much for your effort and research in this interesting topic. I am grateful for your contribution.
Introduction:
- Line 113 -typ error "safe have"
Thank you, we have corrected the typo.
- Lines 138-140: "However, most of the above-mentioned studies utilized exclusively self-report questionnaires, whose answers can be distorted in specific cases [53], and withdrawal is measured by indicator rather than a specifically designed measure" - what did the authors infere here? were the answers distorced? Please be very clear in your statements, do not leave any uncleared inference on reviews and results. If this really happened than just change the sentence.
We thank Reviewer2 to give us the chance to clarify this point. According to the citation, we clarified as follows “whose answers concerning attachment can be distorted in case of partially unaware defensive mechanisms typical of teenagers classified as insecure in interviews, such as idealization or anger towards parents [50]. Moreover, withdrawal is measured by indicator rather than a specifically designed measure.”
Materials and Methods:
- Authors declared to have used SPSS Version 54, this might be an error since the actual version is 28.
We thanks to have note this typo, we have corrected with Version 24.
- A guide / cut-off for the Pearson correlation coefficient is missing. Please inform the reader how to interpret these values.
At the beginning of the “Analytic Plan” we added to the sentence “All analyses were considered statistically significant with p <.05”, this additional information “The size of zero-order Pearson's correlation coefficients (r) has been considered as small (.10), medium (.30), or large (.50).”
- Lines 214-215: authors described the sample and affirm that the gender distribution is 54% boys, but at Table 1 we can see that the frequency of boys in the sample is actually 42% (38 out of 91) - where is the missing piece?
We are sorry, that was a problem with all the numbers in the paper (including references numbers). Now we have checked them entirely and restored the 42% of the percentage of males in the sample as Reviewer2 noted. We did not have missing data because we selected only teenagers that completed both the YSR and the FFI from a larger sample.
- In generally: authors comment on the signification of comparisons based on p-values being higher than different cutoffs, e.g. line 319 p > 0.095 or line 340 p > 0.067 - what are those values? Usually one determines a significance level, say 0.05 and report the significane according to that.
Thank you for this suggestion that helped us to make clearer the results section. As suggested, in case of non significant results, we replaced these information with a more intuitive “p > .05”.
- Is it possible to display (Figure 1) standardized beta coefficients for the models? How are the readers supposed to interpret the a=0.3251 and b=1.534 if they are not at the same scale?
We have modified the figure with the standardized beta coefficients, and specifying in the text where we used standardized values were used.
Kind regards
The Authors
